# A Survey on Recent Advances in Social Robotics

**Karim Youssef, Sherif Said \*, Samer Alkork and Taha Beyrouthy**

College of Engineering and Technology, American University of the Middle East, Egaila 54200, Kuwait; clement.joseph@aum.edu.kw (K.Y.); samer.alkork@aum.edu.kw (S.A.); taha.beyrouthy@aum.edu.kw (T.B.)

\* Correspondence: sherif.said@aum.edu.kw; Tel.: +965-9897-7322

**Abstract:** Over decades, social robotics has evolved as a concept that presently covers different areas of application, and interacts with different domains in technology, education, medicine and others. Today, it is possible to envision social robots in tasks that were not expected years ago, and that is not only due to the evolution of social robots, but also to the evolution of the vision humans have for them. This survey addresses recent advances in social robotics from different perspectives. Different contexts and areas of application of social robots are addressed, as well as modalities of interaction with humans. Different robotic platforms used in social contexts are shown and discussed. Relationships of social robotics with advances in other technological areas are surveyed, and methods and metrics used for the human evaluation of the interaction with robots are presented. The future of social robotics is also envisioned based on surveyed works and from different points of view.

**Keywords:** social robotics; assistive robotics; artificial intelligence; human-robot interaction

## 1. Introduction

A social robot can be defined as a robot designed to operate and interact with humans, in contexts similar to human–human interaction contexts. Knowing that several types of machines and robots can interact with humans in different ways and modalities, a question emerges on what differentiates a social robot from a robot that is not social. In the recent literature, several definitions of social robots can be found, showing differences in visions that authors have of what can be called a social robot. This also arises because there is no agreement on the definition of robots, regardless of their social qualification. The question of defining social robots has been addressed in [1] where scientific and popular definitions of social robots have been examined. The work presented in [1] shows the following points among other findings:

- Not all articles using the "social robot" label defined what the authors meant by it.
- In highly pertinent articles, the majority acknowledges the lack of a generally accepted and accurate definition and the difficulty of defining a social robot.
- There has been an increase in the number of times the label "social robot" has been mentioned since the late 1990s.
- Among the definitions made in scientific articles, the following properties are associated with social robots: autonomy, ability to act in a socially appropriate manner, and in meaningful social interactions, communication, intelligence, operation according to established social and cultural norms, the ability to sense the presence of humans and to engage in physical acknowledgment, the ability to use gestures, to express and/or perceive emotions and to engage in a conversation [2–6].
- Different alternative future-oriented definitions have been made, mentioning a merge between biological and technological elements in social robots, and social robots representing applications of technology meant to solve social problems that are non-technical in nature, for pressing issues in contemporary society [7,8].

Based on the previous definitions and findings obtained from [1], and on other work done in the field of social robotics, it can be stated that social robots can do services and use different means of communication while interacting with humans. It is possible to envision social robots in different types of areas, such as industrial, educational, commercial, and homes, being part of the daily life of humans [9,10]. Social robotics is becoming a major area of research, with the development of new techniques allowing computer systems to process signals and extract information efficiently, such as sound, images, and natural language. Furthermore, social robots can convey information to humans via different means such as speech, gestures, and facial expressions [11]. Advances in machine learning and pattern recognition are allowing robots to acquire realistic social behaviors involving behaviors in speech and gestures [12–17]. This enabled robots to be active conversation parties either in text-based or voice-based conversations. It is possible presently to envision robots in applications that would not have been expected two decades ago, such as voice messaging [18], ecology awareness [19], health data acquisition in hospitals [20] and vocabulary enhancement games [21].

This paper addresses recent advances in social robotics from different points of view. This domain has witnessed a rise in popularity allowing it to expand to various areas of application. Research work has been done with robotic platforms that are either commercially available, programmable, and re-configurable, or designed and manufactured for specific persons and specific usages. Such platforms are equipped with sensing and actuation capacities allowing perceptions of the environment, including humans and other robots, and analysis of the perceptions and actions used in the interaction such as emotion display, gestures, and speech. All this would not have been possible without the advances taking place in several fields, ranging from engineering fields as mentioned previously to psychological fields where the assessment of the perception that humans have of robots plays an important role in their designs and usage. Different works related to all the previously mentioned points are reviewed and cited. The contributions of the proposed survey can be summarized by the following points, each supported by different previous works and studies:

- Demonstrating the extent to which social robotics can be involved in humans' life.
- Proving different capacities allowing a robot to interact in a social manner.
- Emphasizing the impacts of technology and research on social robots.
- Addressing the human side of the human-robot interaction.
- Providing a multi-viewpoint image of the future of social robots.

This survey does not claim to be exhaustive but covers enough references corresponding to each point, allowing a reader who is not specialized in all the mentioned fields, to have a view of the recent work being done in each one. As the aim is to show trends in social robots across different disciplines, works have been included in the survey based on their relevance and date. Robotics journals and robotics conference proceedings addressing social robotics have been relied on to access publications. Mainly, publications from the International Journal of Social Robotics from the last 4 years were accessed if relevant to the survey alongside other sources. A part of the publications was obtained by searching for specific keywords such as "social robot" and "assistive robot" while others were obtained by navigating through the sources. The authors conducting the survey and selecting the articles are researchers in robotics, PhD holders working in a robotics research center with activities and previous publications in the field of social robotics. Figure 1 shows the different points addressed in the survey.

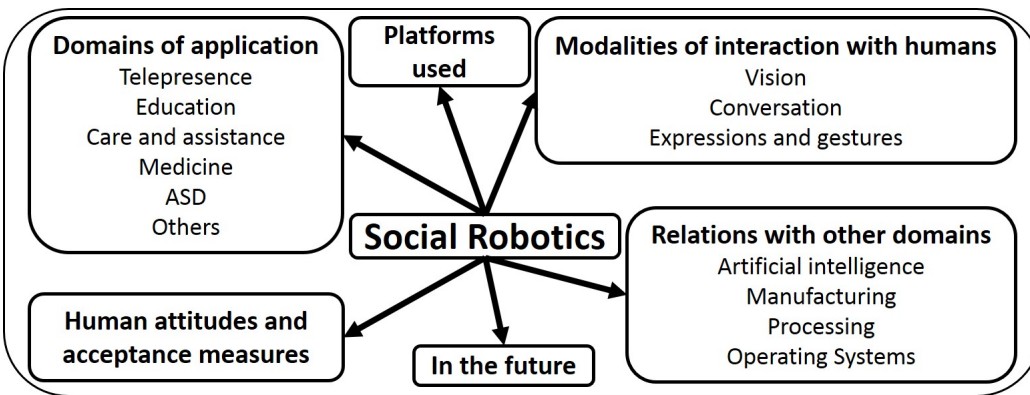

**Figure 1.** Points addressed in the survey.

This paper is organized as follows. Section 2.1 addresses recent advances in social robotics, emphasizing the usages of social robots in different applications. Section 3 addresses modalities of interaction between humans and social robots, mainly showing vision, conversations and gestures. Section 4 shows characteristics of different platforms used in social robotics contexts. Section 5 tackles the advances in different domains, impacting social robotics, namely artificial intelligence, manufacturing technologies, sensing and actuation technologies, and operating systems. In Section 6, different metrics and methods of the measurement of the acceptability of social robot interactions are presented. Later, a link between the past, present, and future applications of social robotics is made in Section 7. Finally, Section 8 concludes the paper.

## 2. Domains of Application

Research works done in social robotics reveal the wide array of domains and applications they are having. Social robots are being proposed in telepresence, medicine, education, entertainment, assistance, and other domains. Benefiting from their information acquisition and processing, and actuation capacities, social robots are conceived to either replace or assist humans in daily social interaction contexts. The following subsections show examples of work where social robots have been used in several domains. Table 1 summarizes several the surveyed works shown in the following subsections. It shows the different applications, the end users, and the robots used in each study. These studies, along with others will be explained further in the next subsections and the robots used will be addressed in more detail in Section 4.

**Table 1.** Examples of applications and experiments where social robots were used, with the respective end users.

| Study | Robot | Research Goal/Application | Targets/End Users |
|---|---|---|---|
| Shiarlis et al. [15] | Teresa | Telepresence-behavior in interaction | Unspecified |
| Shiarlis et al. [22] | Teresa | Telepresence-participation in social events | Elderly |
| Niemelä et al. [23] | Double | Telepresence-communication with family members | Elderly |
| Zhang et al. [24] | Unspecified | Telepresence-Control with eye gaze | Persons with motor disabilities |
| Hood et al. [25] | Nao | Handwriting learning | Children |
| Engwall et al. [26] | Furhat | Second language learning | Various |
| Kanero et al. [27] | Nao | Second language learning | Adults |
| Shimaya et al. [28] | CommU | Communication in lectures | Students and lecturers |
| Reyes et al. [29] | Nao | Assistance in class | Students and lecturers |

**Table 1.** *Cont.*

| Study | Robot | Research Goal/Application | Targets/End Users |
|---|---|---|---|
| Vogt et al. [30] | Nao | Second language learning | Children |
| Schicchi et al. [21] | Pepper | Vocabulary enhancement | Children |
| Obayashi et al. [31] | Mon-chan | Care in nursing homes | Elderly |
| McGinn et al. [31] | Stevie | Care in a care facility | Residents and staff |
| Luperto et al. [32] | Giraff-X | Assistance at home | Elders |
| Ismail et al. [33] | LUCA | Analysis of attention | Children with cognitive impairment |
| van der Putte et al. [20] | Pepper | Data acquisition | Hospitalized patients |
| Schrum et al. [34] | Pepper | Encouraging physical exercise | Dementia patients |
| Moharana et al. [35] | Different robots designed | Designing robots for dementia caregiving | Dementia caregiver support groups |
| Anzalone et al. [36] | Nao | Environment perception | Children with ASD |
| Huijnen et al. [37] | Kaspar | Making contact and catching attention | Children with ASD |
| Taheri et al. [38] | Nao | Body gesture imitation | Children with ASD |
| Chung [39] | Nao | Enhancement of social skills | Children with ASD |
| Striepe et al. [40] | Reeti | Implementing behaviors of a robot storyteller | Persons aged from 18 to 30 |
| Desideri et al. [41] | Nao | Studying gaze aversion in human-robot interaction | Children |
| Filippini et al. [42] | Mio Amico | Assessing the emotional state of robot interlocutor | Children |
| Uluer et al. [43] | Pepper | Assistance for hearing disabilities | Children |
| Castellano et al. [19] | Pepper | Improving attitudes toward recycling | Children |
| lio et al. [44] | ASIMO | Guidance for a science museum | Various, museum visitors |
| Shi et al. [45] | Robovie | Flyer distribution | Various, pedestrians in a shopping mall |

### 2.1. Telepresence

In telepresence applications, a user can rely on a robotic platform to ensure a certain extent of social interaction with other persons while being at a distant location from them. Different technologies have been implemented on social robots to ensure a realistic interaction from both user and interaction partner sides. Telepresence robots require features such as autonomy, controllability, maneuverability, and stability to ensure safe interaction with humans [46]. For instance, in [15], a deep-learning approach has been proposed for a telepresence robot to learn by demonstrating how to maintain an appropriate position and orientation within a group of people, and how to follow moving interaction targets. In this context, the robot has been qualified as semi-autonomous as its pilot still had control over certain high-level tasks. A similar platform has been used in [22] for the interaction between users in and outside an elderly day center. In [46], a robotic telepresence system design and a control approach for social interaction have been presented. The robot has been equipped with capabilities of vision, hearing, speaking, and moving, all controlled remotely by a user. In [23], a study has been shown where a Double telepresence robot was installed in rooms of care homes, for the purpose of allowing old persons to communicate with their family members. Despite some technical difficulties, the experience of using this system was positively evaluated by persons involved in the study. Figure 2 shows a Double telepresence robot [47]. A telepresence robotic system for people with motor disabilities has been proposed in [24]. Eye gaze was used as an input to the system as an

eye-tracking mechanism involving a virtual reality head-mounted display was intended to provide driving commands to the robotic platform.

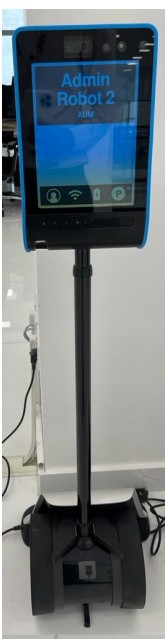

**Figure 2.** A Double telepresence robot in the American University of the Middle east.

### 2.2. Education

Robots have also been involved in education where they had applications in language teaching, teaching assistance, writing, and vocabulary enhancement for example [48–50]. Indeed, they can facilitate learning and improve the educational performance of students, adding social interaction to the learning context in certain cases [51]. In this context, the attitudes of teachers and students towards robots are important to evaluate and the study shown in [52] tackled the negative attitude toward robot scale (NARS) that was developed in [53] to measure general human attitudes towards robots. The study pointed to the importance of knowing the attitudes of teachers towards robots used in classes. Additionally, the study made in [27] studied the acceptability of robots by teaching second language words to adults with the robot and human tutors. A negative attitude toward robots was shown to have a possible negative impact on the ability of individuals to learn vocabulary. Second language tutoring was addressed in [30,54] with children and a social robot with a tablet. Children were not shown to be able to learn more words when learning from a robot and a tablet than from a tablet without a robot. Additionally, iconic gestures from the robot were not shown to help children learn more words. A closely related topic in second language learning was addressed in [26] where different interaction styles of human moderators in language café style conversations were developed for a robot in the role of a host in spoken conversation practice. These styles were rated differently by human participants, due to different factors, not only due to the robot interaction but also due to the participants themselves. This allowed us to state that interaction can be improved, taking these different factors into account. In [21], the humanoid robot Pepper has been used in vocabulary enhancement in children in a game scenario intended to improve their abilities to manipulate and learn words. The capabilities of the Pepper robot such as connecting to the Internet, and acquiring and emitting sound signals have been exploited to accomplish a proper interaction with children in this context. A robotic platform was used to support professors in mathematics classes in [29]. The Nao robot was used for giving theoretical explanations, as well as the instructions and evaluation of the activities made during class. It was programmed before each session to move the most humanly possible to generate accurate visual gestures. Moreover, the vision and sound generation capacities of Nao were exploited for the interaction. In [28], a robotic platform was used for transferring students'

opinions or questions to the lecturer. The desktop humanoid robot was collaboratively controlled and provided a messaging interface where messages consisted of questions or opinions of students to be uttered by the robot. This approach was shown to improve the participation of students during lectures. Another paradigm of learning in children is learning by teaching , which engages a student in the act of teaching another. In [25], a humanoid robot was used as a handwriting partner with simulated handwriting mistakes intentionally made, being typical of children learning to handwrite. Children taught the robot how to write, learning through their teaching.

*2.3. Care and Assistance*

Another domain of application where social robots have emerged is assistance in health and daily care services. In [55,56], the increasing demand for elderly care and the Socially assistive robot (SAR) technology roles in this field are highlighted. It was proposed in [57] that socially assistive robots could support people with health conditions in social interactions, with the aim of improving their health and well-being. For elderly people, social robots can be embedded into their homes or care facilities and play different roles. In [58], the field of in-house assistance for older adults was reviewed. It is suggested that the deployments that have been made for robots for in-house assistance are mostly prototypes and that robots have yet to succeed as personal assistance. It is reported that in healthcare, robots have a variety of applications, and can be classified into three categories: surgical, rehabilitation, and social robots. Furthermore, social robots in this context were divided into service and companion categories, dedicated for assistance in one or more tasks, or for user companionship, respectively. In the last context, the appearance of the robot and the extent to which it resembles a human being were reported to affect its acceptability by end users [59,60]. In this context, the study made in [61] focused on the perception of care robots among end users. It addressed the end users' understandings, assumptions, and expectations of care robots. The study covered different groups of stakeholders such as line managers, frontline care staff, older people, and students training to become careers. Congruent frames between the groups regarding the understanding of the nature of care robots were found. But incongruent frames of the nature of care robots were found between the different groups when addressing the question of sketching the ideal robot. The study identified adequate training, usability, and finances among potential criteria for the successful use of care robots. Perceptions of stakeholders of socially assistive robots were also addressed in [62] where a study on a robot known as Stevie was conducted with older adults and professional care workers in a retirement community. Focus groups were formed where the robot was teleoperated and observations and discussions were made. Staff and residents had different views and concerns about the robot but they both saw its potential utility and suggested many possible use-cases. Older people care has also been addressed in [31] where a user-centered soft and bedside communication robot was developed and evaluated. A collaborative and iterative development process was adopted, involving different stakeholders. The resulting system helped in the improvement of the mood and behavior of participants, as well as in obtaining a positive engagement from their side. Another aspect of assistance was addressed in [63] where an approach to develop a robot for the assistance of workers with intellectual and developmental disabilities was presented.

In the context of care and assistance for older persons, Ambient Assistive Living (AAL) has been defined, as reported in [64] from [65] as "the use of information and communication technologies (ICT) in a person's daily living and working environment to enable them to stay active longer, remain socially connected and live independently into old age". AAL covers tasks such as observation and the detection of events such as falls but goes beyond that to interact with users [64]. In this context, the integration of socially assistive robots into AAL has been shown in [32] to succeed in long-term support to older adults. The authors reported that the robot incentivized the usage of the system but

slightly lowered its overall acceptability. This work used the Giraff-X (a version of the Giraff teleoperated robot [66,67]) as an embodiment for a virtual caregiver at the elder's house.

*2.4. Medicine*

Another field of application of social robots in health care. Different usages can be found for social robots in this context, ranging from the assistance of nurses to rehabilitation [10]. The interventions of socially assistive robots in supporting mental health in children have been reviewed in [68]. It was found that the contexts of the interventions affect their impacts. Indeed, the place, the culture of the user, and the robot used are all factors contributing to the outcomes of the intervention. The study showed different robotic platforms used in this context and reported consistent positive outcomes such as relief of distress and increase of positive effects regardless of the robot used. However, disparities have been seen between outcome measures, robots used, and study quality. The usage of a social robot was shown to have possible benefits in attention improvement for children with cognitive impairment [33]. In this context, a child-robot interaction was designed and implemented, consisting of several modules during which the children played short games with the robot, taking the capacities of the robot into account. Additionally, dementia was addressed in [34,35]. The research work in [35] focused on designing robots for dementia caregiving, addressing the needs of both the caregiver and the caregiver. This covered the intended purpose and functions of the robots as "robots for joy", "robots for repetition" and "robots for wellness" were designed. Additionally, different morphologies and modalities for interacting with the robots such as voice interaction where the voices of people that whom caregivers were familiar were discussed. Moreover, different roles were assigned to robots, such as "the bad guy", "the facilitator" and "the counselor". The Softbank robot, Pepper, was used in [34] for encouraging exercise in dementia patients. Specifically, the study used simple dance moves as the exercise modality due to the engagement and repetitiveness of dancing. A heart-rate monitor was used for sending feedback to the robot to adjust the intensity of the exercise. Preliminary results were reported to be promising. Pepper was also used in [43] in a system developed for the audiometry tests and rehabilitation of children with hearing disabilities. Positive and negative emotions of children were shown to be better distinguished when they interact with the robot than in setups without the robot. Social anxiety disorder, a condition pushing people to fear social situations, was addressed in [69] where an overview of certain usages of social robots in clinical interventions was made. This work proposed that social robots can be used to complement the work of clinicians. Additionally, Pepper's usage in health data acquisition was explored in [20] to act as a nurse assistant and reduce data registration workloads on nurses.A multimodal dialogue involving verbal, gesture, and screen display aspects were designed to facilitate the interaction between the robot and the patient. Evaluations made by patients and nurses showed the possible acceptability of the robot. Another usage of robots was shown in [56], not directly as a health assistant, but as an assistant in enhancing the skills of nursing students, specifically in patient transfer, where a patient is moved from a bed to a wheelchair, and vice-versa. The robot in this work simulated a patient to be transferred while measuring different motion parameters during the transfer and assessing whether the transfer was made accurately by the nursing student or not. Indeed, proper body mechanics need to be used in this task, indispensable to the patient's daily life, especially with elderly patients affected by weaknesses in their limbs. Results showed that the robot can be a good substitute for an actual patient when performing this task. Rehabilitation was addressed in [70] and specifically, the trust of users interacting with a rehabilitation robot. Exercises were performed at different velocities of robot motion and data on participants' heart rates and perception of safety were collected. Notably, the perception of safety was negatively affected by increasing velocity and exercise extent. Another application of socially assistive robots has been shown in [71] where robot prototypes that assist persons in sorting their medications have been developed and tested, to organize the days and times pills should be taken.

## 2.5. Autism Spectrum Disorders

Autism Spectrum Disorders (ASD) cause abnormalities of impaired development in social communication and interaction, and in restricted and repetitive patterns of behavior, interests, or activities. Individuals with ASD have difficulties interacting and communicating with others [72,73]. This is due to their inability to understand social cues and the behaviors and feelings of others. Research on information communication technology (ICT) has been active in the domain of the education of people with autism [74]. In the same context. different works in socially assistive robotics have tackled the treatment of individuals with ASD, increasingly since 2000, with different directions of research [75]. Such works target the improvement of the social functioning of children with ASD [39]. One of these directions is imitation, as a deficit in imitation is a symptom of ASD. The study made in [38] compared the body gesture imitation performance of participants with ASD and typically developing subjects. It also compared this performance in robot-child and adult-child imitation tasks. In the presented experimental setup, both participants with typical development and with ASD performed better in adult-child mode than in robot-child mode. Additionally, participants with typical development showed better performance than participants with ASD. Additionally, in [36], children with typical development and with ASD performed better with a therapist in a joint attention elicitation task. Joint attention is related to social relationships and has been defined in [76] as a triadic interaction between two agents focusing on a single object. Additionally, in [36], children with typical development performed better than children with ASD with the robot. Another direction of research, where the usage of robots proved efficient in the enhancement of social skills of children with ASD was shown in [39]. In this study, a robot was used for an intervention in a social skill training program that consisted of different phases with and without the robot. Results showed that the intervention of the robot improved the social motivation and skills of children with ASD, measured by eye contact frequency and duration, and verbal initiation frequency. This improvement lasted even after the robot was withdrawn from the program. A similar result was obtained in [37] where children with ASD participated in sessions divided into sessions with the Kaspar robot and sessions with a human teacher. The usage of the robot increased interactions among children such as non-verbal imitation, touching, and attention duration. The Kaspar robot's evolution is shown in [77]. Initially, the Kaspar robot was developed in 2005 for research in human-robot interaction, then it was adopted for investigation as a therapeutic device for children with ASD which has been its primary application and target of improvement since then. The evolution of this robot benefited from the knowledge improvement in the therapy of children with ASD and shows hardware modifications and improvement of sensory devices aiming to improve its usability and autonomy for child-robot interaction.

## 2.6. Other Applications of Children Companionship

As reported, social robots have been used with children in applications such as education and ASD. Nevertheless, other applications of social robots used with children can be reported from other domains such as entertainment, awareness-raising and cognition, perception, and behavioral studies. In [19], a social robot was used in a game intended to make children more aware of the importance of waste recycling. The game had 2 players: the child and the Softbank robot Pepper, and a human judge. The study reported promising results in changing children's attitudes toward recycling and showed a positive evaluation of Pepper by children. Children's gaze aversion was addressed in [41] where gaze aversion was reported from other sources to refer to human being reflexive redirection of the gaze away from a potentially distracting visual stimulus while solving a mentally demanding task, facilitating thinking. The study evaluated the influence of the interaction with a humanoid robot on children's gaze aversion. Results showed that gaze aversion rates increased when children interacted with other humans, or with robots that were told to be human-controlled, in contrast with their interactions with robots controlled by computers. These findings were linked to the perception children make of minds in their interaction

agents. Child-robot interaction was also explored in [42] to develop a method for emotion recognition relying on functional infrared imaging. It allowed for the assessment of the level of child engagement while interacting with an artificial agent and the presented work was said to constitute a step toward a more natural interaction between a child and an artificial agent, based on physiological signals. In [40], the focus was on implementing the behavior of a robot storyteller using an analysis of human storytellers. The effects of implementing emotions in the storytelling, contextual storyteller head movements, and voice fitting to characters were evaluated. Positive impacts on listeners were found in emotional robot and voice-acting robot storytellers. Contextual head movements, on the other hand, did not have effects on the perception users make of the robot storyteller. The usage of robots in childcare was addressed in [78] where the requirements, needs, and attitudes of working parents toward childcare social robots were identified. The study suggested socialization, education, entertainment, and expert counseling as childcare functions of social robots and created questionnaire items to explore different aspects of the parents' views of these different functions. The results suggested positive impacts of social robots in childcare through aspects such as social interactions and entertainment. Different parenting conditions such as parenting styles (work-oriented, dominant, etc.) and children's ages were reported to change parents' needs for specific childcare functions. This implies that robots can be strategically designed and introduced to customer groups in line with their characteristics. The study shown in [79] focused on games involving humans and robots in physical and demanding activities. In this context, robots need to be perceived as rational agents aiming to win the game, and the study focused on deciding and communicating deceptive behaviors in robots. This strategy improved human-robot interaction by helping robots match the expectation of interacting with people to attribute rationality to the robot companion. Another field of research where social robotics had applications in affective computing, aiming to understand the effect of a person using specific signals and modalities, and applied in education for example [80]. In [81], a children companion robot was equipped with the capacity of real-time affective computing, allowing the robot to adapt its behavior to the effect of the child it is interacting with, improving the interaction.

### 2.7. Other Domains of Research and Application

As stated in [82], human-robot interaction has been explored in children, adults, and seniors, but it was less explored in teens. The authors in [82] state that designing robots for interaction with teens is different from other types of human-robot interaction. Additionally, aside from the different domains of research and application that have already been shown, social robots have been used and explored in different contexts, for different objectives. For instance, "edutainment", where robots participate in people's education and entertainment can be mentioned [83]. Additionally, several studies have been made to improve human-robot interaction by embedding human social skills in robots. For example, instead of using pre-programmed manually crafted gestures, a humanoid robot learned, using neural networks and a database of TED talks, to generate gestures by the uttered speech as humans would, in [12]. Storytelling is also a field of human-robot interaction where different aspects of robot behavior can be explored [84] a service robot's ability to adapt its behavior was addressed by implementing a human-like thought process. Behavior in this context can be defined as a combination of a facial expression, a gesture, and a movement. Social intelligence and familiarity with robots have also been the objective in [14,85]. In [86], A robot was equipped with the ability to assess whether the human interaction partner is lying or not, in the purpose of assessing his trustworthiness and improving the interaction. In [14], a robot used deep neural networks to learn human behavior based on data it gathered during its interactions. The purpose was to use the most appropriate action among waving, looking toward human, waving, and handshaking Additionally, in the context of social intelligence, a vision-based framework for allowing robots to recognize and respond to hand waving gestures were presented in [16], increasing its social believability. Furthermore, a humanoid robot was endowed with human-like welcoming behaviors with

enthusiasm for the purpose of drawing the attention of persons entering a building in [17]. In a related application in terms of constraints, a flyer distributing robot for pedestrians in a shopping mall has been presented in [45]. Indeed, the robot needed to draw the attention of pedestrians, plan its motions and behave in a manner helping to make the pedestrians accept the flyers. Additionally, a guide robot was developed in [44] for a science museum. The robot had abilities to build relationships with humans through friendly attitudes and was positively evaluated by visitors. Finally, a design and framework were shown in [87] for a robot intended to be used as a receptionist in a university. The platform consisted of an animatronic head with several degrees of freedom and a capacity in engaging in conversations without *'a priori* information about questions it may have to answer. Such an application is an example of how a social robot can combine aspects from design, hardware, software, artificial intelligence, and communication to play roles that are usually attributed to humans.

## 3. Modalities of Human-Robot Interaction

As in human–human interaction, several modalities can be used at once in human-robot interaction in social contexts. Vision, eye gaze, verbal dialogue, touch, and gestures are examples of modalities that can be used in this context. In a social context, the intelligence that a robot display depends on the modalities it uses and each modality can have specific importance and effect on the human side of the interaction [88] which translates into the degree of trust that the robot has [89]. Moreover, the acceptance of robots in social interaction depends on their ability to express emotions and they require a proper design of emotional expressions to improve their likability and believability [90] as multimodal interaction can enhance the engagement [91–93].

### 3.1. Vision Systems in Robots

Visual perception provides what was suggested to be the most important information to robots, allowing them to achieve successful interaction with human partners [94]. This information can be used in a variety of tasks, such as navigation, obstacle avoidance, detection, understanding, and manipulation of objects, and assigning meanings to a visual configuration of a scene [95,96]. More specifically, the vision has been used for the estimation of the 3D position and orientation of a user in an environment [97], the estimation of distances between a robot and users [98], tracking human targets and obtaining their poses [83], understanding human behavior aiming to contribute to the cohabitation between assistive robots and humans [99]. Similarly, the vision has been used in a variety of other applications, such as recognizing patterns and figures in exercises in a teaching assistance context in a high school [29], detecting and classifying waste material as a child would do [19], and detecting people entering a building for a possible interaction [17]. Moreover, the vision has been used in [71] for medication sorting, taking into account pill types and numbers, in [100] for sign recognition in a sign tutoring task with deaf or hard of hearing children, and in [101] as part of a platform used for cognitive stimulation in elderly users with mild cognitive impairments.

### 3.2. Conversational Systems in Robots

Although some applications of social robotics involve robots taking vocal commands without generating a vocal reply [102], interactions can be made richer when the robot can engage in conversations. A typical social robot with autonomous conversation ability must have the capacity to acquire sound signals, process them to recognize the speech, recognize the whole sequence of words pronounced by the human interlocutor, formulate an appropriate reply, and synthesize the sound signal corresponding to the reply, then emit this signal using a loudspeaker. The core component of this ability is the recognition of word sequences and the generation of reply sequences [103]. This can rely on a learning stage where the system acquires the experience of answering word sequences by observing a certain number of conversations that are mainly between humans. Techniques used in

this area involve word and character embeddings, and learning through recurrent neural network (RNN) architectures, long short-term memory networks (LSTM), and gated recurrent units (GRU) [103,104]. It is to note that not all social robotic systems with conversational capacities have the same levels of complexity as some use limited vocabularies in their verbal dialogues. In this context, Conversation scenarios were seen in [31], verbal dialogue in [20], dialogues between children and a robot in [33], and some word utterances in [19]. A review of conversational systems usages in psychiatry was made in [105]. It covered different aspects such as therapy bots, avatars, and intelligent animal-like robots. Additionally, an algorithm for dialogue management has been proposed in [106] for social robots and conversational agents. It is aimed at ensuring a rich and interesting conversation with users. Furthermore, robot rejection of human commands has been addressed in [107] with aspects such as how rejections can be phrased by the robot. GPT-3 [108] has emerged as a language model with potential applications in conversational systems and social robotics [109]. However, in several conversational systems, problems have been reported, such as hallucinations [110,111], response blandness, and incoherence [103]. The research work presented in [87] aimed at improving the conversational capabilities of a social robot by reducing the possibility of problems as described above, and improving the human-robot interaction with an expressive face. It intended to have a 3-D printed animatronic robotics head with an eye mechanism, a jaw mechanism, and a head mechanism. The three mechanisms are designed to be driven by servo motors to actuate the head synchronously with the audio output. The robotics head design is optimized to fit microphones, cameras, and speakers. The robotics head is envisioned to meet students and visitors in a university. To ensure the appropriateness of the interactions, several stages will be included in the control framework of the robot and a database of human–human conversations will be built upon for the machine learning of the system. This database will be built in the aim of training the conversational system in contexts similar to its contexts of usage. This will increase the adequacy of the conversational system's parameters with respect to the tasks it is required to do, and increase the coherence and consistency of the utterances it produces. For that, the recorded data will comply with the following specifications:

- Context of the interactions: Visitors approach the receptionist and engage in conversations in English. Both questions and answers will be included in the database.
- Audio recordings of the conversations: a text by speech recognition modules is used to transcript the conversation.
- Video recordings of the interaction, showing the face and upper body of the receptionist, with a quality of images usable by body posture recognition systems.
- The collected data will be used to progressively train the system. Each conversation will be labeled with the corresponding date, time and interaction parties.
- Participants will be asked to be free to ask questions they may have to inquire about the center in English, without having any other constraint or any specific text to pronounce.

*3.3. Expressions and Gestures*

Aside from the ability to process and generate sequences of words, a social robot requires more capacities to increase engagement and realism in the interaction with a human. This can be done through speech-accompanying gestures and facial expressions. Indeed, facial expression has an important role in communication between humans because it is rich in information, together with gestures and sound [112–114]. This issue has been studied in psychology, and research indicates that there are six main emotions associated with distinctive facial expressions [115]. At Columbia University [116], scientists and engineers developed a robot that can raise eyebrows, smile, and have forehead wrinkles similar to humans. This robot can express the face more accurately compared to the rest of the robots. This robot, called Eva, can mimic head movements and facial expressions. In this robot, 25 muscles are used, and 12 of them are dedicated specifically to the face. These muscles can produce facial skin excitations of up to 15 mm. In other works, different examples can be found for applications of gestures and expressions in social robotics. For instance,

gestures have been combined with verbal dialogue and screen display in [20] for health data acquisition in hospitals with Pepper. In [40], a robot with the ability to display facial expressions was used in studies related to storytelling robots. These studies focused on the roles of the emotional facial display, contextual head movements, and voice acting. In [113], a framework for generating robot behaviors using speech, gestures, and facial expressions was proposed, to improve the expressiveness of a robot in interaction with humans. of interaction with human users

## 4. Robotic Platforms Used in Social Robotics

Over the different research works covered in this paper, different robotic platforms have been used, with different possible modalities of interaction with humans. It can be easily stated that the Softbank Robotics [117] Nao robot [118] has had a great share of the previous work and studies [12,25,27,29,30,36,38,39,41,48,50,54,119,120] for the different aspects of its usability as it will be shown later. Another humanoid robot of Softbank Robotics, Pepper [121], was also widely used [16,19–21,43]. Other robotic platforms that can be mentioned are ASIMO [44,84,122,123], *iCub* [124], TERESA [15,22], Double [23,47], Furhat [26,125], Stevie [62,126], LUCA [33] which was inspired by the OPSORO platform [127,128], The Universal Robot Trainer (URT) [70], Kaspar [37,77], Mio Amico [42], Reeti [40,129], BabeBay [81], Kiddo [49], Paro [130,131], Tega [132], and Huggable [133,134]. Additionally, Sophia has emerged as a framework for robotics and artificial intelligence research that made appearances in television shows and conferences, also becoming the world's first robot citizen and the first robot Innovation Ambassador for the United Nations Development Program [135]. It is a human-like robot with the capacity to dialogue, show expressions, walk, shake hands and perform gestures [136]. Another platform for social robotics, enabling research and development applications and providing a wide range of capabilities such as appearance and voice customization, human-mimicking expressions, user tracking and conversations is Furhat [125,137]. Additionally, Sony Aibo has been presented in 1999 as a dog-like robot and is still being produced in its fourth generation currently [138,139]. The research was conducted on the social behaviors toward Aibo and the bonding humans can have with it [140].

Naturally, a common ability of these different types of robotic platforms is interaction with humans. However, the interaction modalities are not always the same, which affects their usability according to the purposes of each study. Figure 3 shows images of the Nao and Pepper robots, exhibiting some of their capabilities such as changing body posture and displaying information.

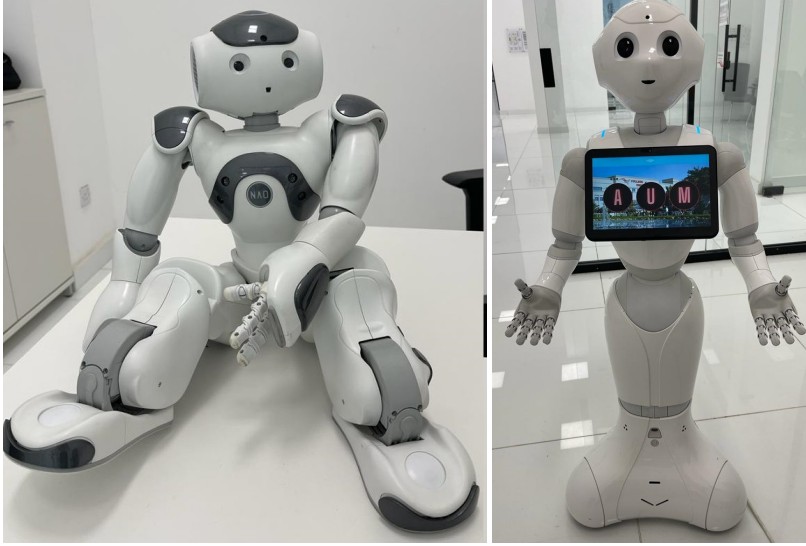

**Figure 3.** Softbank Robotics Nao (**left**) and Pepper (**right**) robots at the American University of the Middle East.

Table 2 shows several platforms with some of their characteristics and features. It allows us to see the different shapes, sizes, shapes, and capabilities of robots used in social contexts. Many robots had different versions since their first appearance, with features and degrees of freedom being added or omitted. In this context, criteria for the development of robots for interaction with humans have been becoming defined and improved over the years. For instance, as stated in [77], a humanoid robot developed for children with ASD should be user-focused rather than technology-focused, usable, reliable in terms of the usage made by its users, and safe. Another parameter to take into consideration when characterizing social robots is the degree of autonomy a robot has, with the modalities of interaction with humans that its capacities allow it to use.

**Table 2.** Examples of robots used in social contexts and some of their characteristics.

| Robot | Appearance | Height (cm) | D.o.F. | Features |
|---|---|---|---|---|
| Nao [118,141] | Humanoid | 58 | 25 | Touch sensors, directional microphones and speakers, 2D cameras, embedded speech recognition and dialogue, programmable, etc. |
| Pepper [121] | Humanoid | 120 | 20 | Touch sensors and microphones and speakers, 2D and 3D cameras, embedded speech recognition and dialogue, programmable, etc. |
| Asimo [142,143] | Humanoid | 130 | 57 | Different proprioceptive and exteroceptive sensors for motion tracking, obstacle detection, image and sound acquisition, etc. |
| Kaspar [77] | Humanoid | 55 | 22 | Color camera, Kinect and IMU sensor, semi-autonomous, Wi-Fi/Ethernet connection. |
| TERESA [15] | Unspecified | Unspecified | Unspecified | semi-autonomous navigation, different proprioceptive and exteroceptive sensors for motion tracking, obstacle detection, image and sound acquisition, etc. |
| Furhat [125] | Human-like face | 41 | 3 | Onboard camera and microphones, speech recognition and synthesis, eye contact, etc. |
| Sophia [135] | Humanoid | 167 | 83 | several cameras, audio localization array, complex and emotional expressions, natural language processing, visual tracking, etc. |
| Giraff [144] | Telepresence | Unspecified | Unspecified | LCD screen, remote control, provides audio and visual cues to the user, data collection for health professionals, etc. |
| Paro [130,131] | Animal (Seal) | Unspecified | Unspecified | Different kinds of sensors, learning to behave as the user prefers, can move its head and legs, etc. |
| Tega [132,145] | Unspecified | 34.54 | 5 | Microphone, camera and accelerometer sensors, autonomous or remote operation, ability to generate behaviors and facial expressions, etc. |
| Huggable [133,146] | Teddy bear | N/A | 12 | Can perceive physical touch, different other sensors, controlled by an application on a smart phone, teleoperation interface, etc. |
| Aibo [138,139] | Dog-like | 29.3 | 22 | Different sensors, cameras, microphones, can recognize faces and voices, capable of simultaneous localization and mapping, etc. |

## 5. Relationships with Recent Advances in Different Domains

The rise of popularity of social robots has, with no doubt, happen not only due to advances in different areas of science and technology but also to the rising acceptability of humans for machines and robots in a variety of environments and contexts. To make a robot, several components are required, including sensors, actuators, processors, communication devices, and chassis materials. Software and data processing techniques are then required

for the control of the different parts of the robot, according to the goals and tasks defined. Advances in different domains, positively impact the usage of social robots, are reviewed in this section.

### 5.1. Artificial Intelligence

Artificial intelligence advances have been benefiting technology on different levels, whether robotic or not [147,148]. In particular, the different tasks required in a social robotic platform, involving aspects of artificial intelligence, have taken benefited from the development and usage of databases, toolboxes and programming environments that have been shown to be reliable and used for that in several works targeting different applications. The increasing availability of machine learning and data processing toolboxes, which are open source in a variety of cases, is allowing fast advances in different areas with applications in social robotics. For instance, OpenPose allows the estimation of body and face characteristics of humans in interaction with robots [12,17,149]. Also, GloVe is a reliable word embedding model used in conversational systems [149,150]. Another example is Handtrack, an end-to-end trainable CNN model that learns to track the human hand [16,151]. Also, studies have been made in the aim of modeling and digitizing human emotions, which can have projections on social robot behavior and intelligence perception [152,153].

### 5.2. Manufacturing Technologies

Robotics in general, and the social robotics domain in particular, have been positively impacted by the advances in technologies related to manufacturing. Indeed, it has been made easier to design, prototype, and manufacture a robotic platform chassis and exterior due to the rising availability of laser cutting, 3D printers, and design software [128,154–157]. Moreover, the advances in semiconductor technologies have made it possible to equip robots with processors and sensors of higher performance, smaller dimensions, and lower costs.

#### 5.2.1. Additive Manufacturing

In [158], the flexibility and customization enabled by 3D printing are highlighted and it is suggested that this technology will continue to be adopted by robotics companies. The same paper reports that even though 3D printing was primarily used for prototyping, it was becoming ready to be used in production. In [159], a literature review of 3D printing in humanoid and social robotics has been made. It leads to a set of design principles contributing to a home-built 3D printed humanoid robot. InMoov, an open-source 3D-printed humanoid robot, has been conceived as a platform for different uses, which is easily reproducible. It was relied on to reach other platforms with the aim of reproducibility, affordability, adding more features, and improvement [160–162]. The work shown in [163] addresses specific polymeric materials used in 3D printing according to their suitability for robotic applications and constraints. Presently, 3D printing-based robots can be seen in robotics education [155], allowing the reduction of costs and demonstration and application concepts, and in entertainment and assistance for children and elderly people [156]. They can also be seen in conversational humanoid robotics [157] allowing the configuration of a human-like face with emotions displayed due to servomotors. The flexibility allowed by 3D printing can also be exploited to easily change designs and reconfigure robots, as shown in [164] where an approach for socially interactive robots reconfiguration for end users has been presented, offering different functionalities. In the same context, 3D printing was relied on in [165] to reach a robot head in an iterative process taking into account the opinions of novices and expert robot users. Additionally, the design shown in [87] was conceived for 3D printing, it consisted of a head shape with several mechanisms inside to actuate the head, the eyes, the jaw, and the neck. This flexibility allows implementing designs with the ability to perform rich interactions with humans through software and hardware-based expression generation. Figure 4 shows this design.

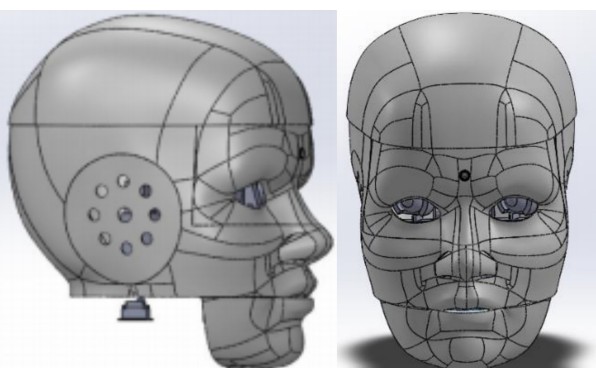

**Figure 4.** Robot head designed for 3D printing with rich expressiveness. Reproduced with permission from [K Youssef, S Said, T Beyrouthy, S Alkork], [BioSMART] ; published by [IEEE], [2022] [87].

### 5.2.2. Semiconductor Devices

Being used in several parts of a robot, semiconductor-based devices have benefited robotic technologies, allowing the expansion of the capabilities and usefulness of robots. At the same time, the usage of robotic technologies in semiconductor manufacturing has allowed remarkable advances in efficiency, productivity and quality [166,167]. The work shown in [168] presents a robotic system architecture, where the role of semiconductors in some of its main components is crucial as follows:

- Processors: to carry out and coordinate different tasks of the robotic system.
- Human machine interface: in screen displays, LEDs, microphones, speakers and their drivers.
- Sensors: for position, speed, current, distance and orientation for example.
- Drivers: for the different types of actuators used in robots.

### 5.3. Processing Technologies

As expectations from social robots evolve, tasks assigned to them increase in computational complexity and require processors, storage spaces, and interfaces capable of handling them. This can be said about various signal and image processing algorithms, machine learning techniques, kinematics, motor command and control, and access to a variety of sensors that the robots can be equipped with. Social robotics has thus taken benefit of the development of processing technologies that are embeddable and reliable in such contexts with real-time constraints. Table 3 lists several robotic platforms and shows their respective processors and other resources. As it can be seen, several platforms are equipped with computational capabilities of average to high-performance computers, with advanced central processing units, spacious memory units, and graphical processing units (GPUs) in some cases. In 2011, an article published on the Robotics Business Review website reported evidence suggesting that GPU processing was already impacting robotics [169]. The article showed the usages of GPUs in autonomous driving, creating depth maps for robots allowing them to navigate and climb stairs for example. In social robotics, and in the same context, GPUs have been used in social robotic experiments, even if not embedded on the robotic platforms [170,171]. For instance, the musical robot companion Shimi, was run in [172] using a GPU for the purpose of enabling it to perform gestures and musical responses to human speech. In this context, NVIDIA developed the Isaac SDK to enable GPU-accelerated algorithms and deep neural networks, and machine learning workflows [173] to accelerate robot developments.

**Table 3.** Examples of social robots and some of their processing characteristics.

| Robot | Processor | Processor Features | RAM | GPU |
| --- | --- | --- | --- | --- |
| Nao V5 & V4 [174] | ATOM Z530 | 1.6 GHZ clock speed | 1 GB | None |
| Pepper V1.6 [175] | ATOM E3845 | 1.91 GHZ clock speed, quadcore | 4 GB DDR3 | None |
| Sophia [176] | Intel i7 | 3 GHZ | 32 GB | integrated GPU |
| ARI [177] | Intel i5/i7 | Unspecified | 8 GB/16 GB | NVIDIA Jetson TX2 |
| QTrobot [178] | Intel NUC i7 | Unspecified | 16 GB | None |
| Furhat [179] | Intel i5 | up to 3.4 GHz | 8 GB | None |
| Giraff [144] | Intel i7 | Unspecified | 8 GB | NVIDIA Jetson TX2 in [32] |
| Asimo [122] | Unspecified | | | |
| Huggable [146] | Computational power of an Android phone | | | |
| Shimi in [172] | ARM | Quadcore | 8 GB | NVIDIA Jetson TX2 |
| Aibo [139] | Qualcomm Snapdragon 820 | 64-bit Quadcore | 4 GB | None |

### 5.4. Operating Systems and Other Software

Aside from the increasing availability of open-source artificial intelligence tools used in social robotics, other types of software and operating systems have been relied on, in allowing social robots operation and tasks. For instance, the Robot Operating System (ROS) [180] is used to build applications in several areas of robotics, including social robots [181]. Additionally, among the various types and usages of robots displayed on the ROS website, social robots such as QTrobot and ARI can be found [182,183] ARI uses ROS to implement capabilities such as perception, cognition, navigation, and expression through gestures and behaviors, and QTrobot uses ROS for emotion, speech, and gestures. ROS has also been used in ROSBOT [184], a low-cost robotic platform with the capacities of obstacle avoiding navigation, face detection, and other social behaviors. Moreover, ROS was used in the implementation of cognitive mechanisms to allow robots to socially interact with humans [185–187]. Figure 5 shows another platform, the UXA-90 humanoid robot, developed to serve several purposes with 23 degrees of freedom and supporting ROS [188].

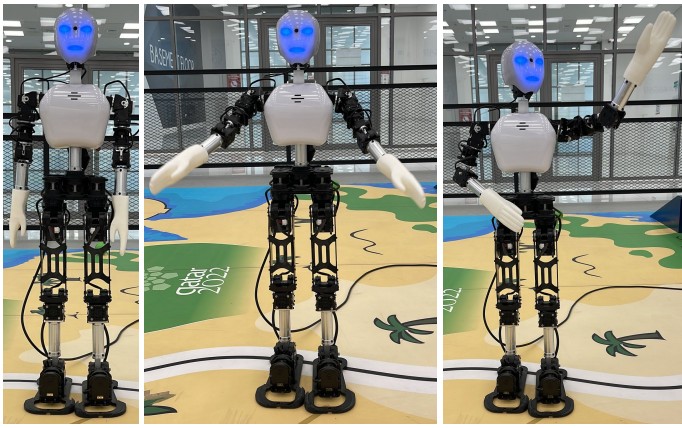

**Figure 5.** The humanoid robots UXA-90 with different body postures at the American University of the Middle East.

The Softbank Robotics NAO and Pepper robots use the NaoQI framework [189], relying on OpenNao which is a GNU/Linux distribution based on Gentoo [190]. However, the ROS driver naoqi_driver can be used for these robots. A review of the most famous robotic frameworks and middleware has been made in [191]. Not all these frameworks and middleware have been witnessed to be used in social contexts, but the paper provides a comparison between them. This comparison took into account metrics such as the operating systems supported, the programming languages that can be used, open-source

aspects, distributed execution, and real-time orientation. ROS has been said to be the robotic framework with the larger momentum and appeal to the community.

Not only have social robotic platforms made use of the available software tools and operating systems but also some tools have been proposed specifically for the social robotics field. In [192], a set of conventions and standard interfaces for HRI scenarios have been presented, designed to be used with ROS, under the ROS4HRI framework. They cover functionalities required for human-robot interaction such as skeleton tracking, face recognition, and natural language processing. The authors of this paper reported the packages people and cob_people_perception as two ROS projects in the human-robot interaction context, which were not multimodal [193,194]. The Human-Robot Interaction Operating System has been proposed in [195] as an infrastructure providing a structured software framework allowing humans and robots to communicate and work as partners, coordinating their actions through dialogue. Additionally, Adaptive Character of Thought-Rational/Embodied (ACT-R/E) has been proposed for human-robot interaction in building models of people to understand how they think [196]. These models were proposed to be used in improving the abilities of robots in interacting with humans.

## 6. Metrics of Human Perception and Acceptability

The usage of social robots in the different environments and contexts presented above is subjected to their acceptability by humans as partners in the interaction. Indeed, to be accepted in social contexts, robots need to show degrees of intelligence, morphology, or usefulness that can be judged positively by users, not to mention cultural influences on expectations towards and responses to social robots [197]. The study published in 2021 in [198] focused on the perception that humans have of the cognitive and affective abilities of robots and began with the hypothesis that this perception varied in accordance with the degree of human-likeness that robots have. However, the results obtained with students on four robots used in the study did not prove this hypothesis. A study made in 2005 in [199] showed the acceptability of persons for robots as companions in the home, more as assistants, machines, or servants than as a friend. More recently, the literature review and study made in [200] mentions anthropomorphism, animacy, likeability, perceived intelligence, and perceived safety as five key concepts in human-robot interaction. The study also emphasized the importance of being aware of human perception and cognition measures developed by psychologists for engineers developing robots. Additionally, according to the tasks expected from the robots, different measures of performance can be made, such as true recognition measures in speech recognition tasks. But a robot can have a high performance in a specific task, without having a positive impact on its social context. Therefore, the performances of robots in social usage are in many cases measured through evaluations made by humans using questionnaires and metrics calculated based on them. The outcomes of such evaluations are affected by the subjectivity of the persons participating in them and their numbers. In this context, certain metrics/measures can be mentioned as follows:

- in [15], a robotic platform was equipped with the capacity to perform the two tasks of group interaction, where it had to maintain an appropriate position and orientation in a group, and the person following. The human evaluation began with a briefing of 15 subjects about the purpose of each task, followed by a calibration step where the subjects were shown human-level performance in each task, followed by interaction with the robotic platform for each task. Then, the subjects were asked to rate the social performance of the platform with a number from 1 to 10 where 10 was human-level performance. The authors suggested increasing the number of subjects and a more detailed questionnaire to be necessary for reaching definitive conclusions.
- the "Godspeed" series of questionnaires has been proposed in [200] to help creators of robots in the robot development process. Five questionnaires using 5-point scales address the anthropomorphism, animacy, likeability, perceived intelligence, and perceived safety of robots. For example, in the anthropomorphism questionnaire (Godspeed I), participants are asked to rate their impressions of the robot with an integer

from fake (1) to natural (5), and from machine-like (1) to human-like (5), and from arti­ficial (1) to lifelike (5). Also in the animacy questionnaire (Godspeed II), participants can rate the robot for example from dead (1) to alive (5), from stagnant (1) to lively (5), and from inert (1) to interactive (5). The authors in [200] report cultural backgrounds, prior experiences with robots, and personality to be among the factors affecting the measurements made in such questionnaires. Furthermore, the perceptions of humans are unstable as their expectations and knowledge change with the increase of their experiences with robots. This means, for the authors in [200], that repeating the same experiment after a long duration of time would yield different results.

- in the context of elderly care and assistance, the Almere model was proposed in [201] as an adaptation and theoretical extension of the Unified Theory of Acceptance and Use of Technology (UTAUT) questionnaire [202]. Questionnaire items in the Almere model were adapted from the UTAUT questionnaire to fit the context of assistive robot technology and address elderly users in a care home. Different constructs are adopted and defined and questionnaires related to them, respectively. This resulted in constructs such as the users' attitude towards the technology their intention to use it, their perceived enjoyment, perceived ease of use, perceived sociability and usefulness, social influence and presence, and trust. Experiments made on the model consisted of a data collection instrument with different questionnaire items on a 5-point Likert-type scale ranging from 1 to 5, corresponding to statements ranging from "totally disagree" to "totally agree", respectively.

Other metrics and approaches for the evaluation of the engagement in the interaction between humans and robots have been proposed. The work presented in [203] proposes metrics that can be easily retrieved from off-the-shelf sensors, by static and dynamic analysis of body posture, head movements and gaze of the human interaction partner.

The work made in [200] revealed two important points related to the assessment of human-robot interaction: the need for a standardized measurement tool and the effects of user background and time on the measurements. The authors also invited psychologists to contribute to the development of the questionnaires. These issues can have implications for social robotics studies that should be addressed to improve assessment quality and results and advance robotic system designs and tasks accurately. More recently, the work shown in [204] proposed a standardized process for choosing and using scales and questionnaires used in human-robot interaction. For instance, the authors in [204] specified that a scale cannot be trusted in a certain study if not already validated in a similar study and that scales can be unfit or have limitations concerning a specific study. In such a case, they should be modified and re-validated.

## 7. Future of Social Robotics

Across the different areas of research that have been shown in Section 2.1, research work has been made in many cases as propositions and proofs of concept with expectations and propositions for future work. Different works have addressed the future of social robotics from their respective contexts and points of view. Additionally, while some suggestions take into account purely technical aspects, such as adding or improving functionalities of robots, others address the importance of robot acceptability and familiarity, as well as the robot users' viewpoints as factors to be taken into account. It is possible to envision robots being used in various everyday life scenarios, from entertainment to education and medical care, engaging in active and rich interactions with humans. Humans would have a clear idea of the capabilities of the robots they interact with, they would attribute degrees of trust to them and grow familiar with their shapes. Robots would be capable of processing various types of signals to understand the needs, behaviors, and emotions of people, and they would convey signals in real time, according to their perceptions. Below are summaries of future expectations from papers in different areas.

- Education: in [28], the authors suggested motivating schoolteachers to introduce a collaborative robot into their lectures. Enabling robot-supported language learning

for preschool children was proposed as a long-term goal in [50]. In [51], where a review of robots in education was made, improving the speech understanding capabilities of robots and reproducing human-like behavior were proposed. In [27], the authors specified that second language anxiety and negative attitudes toward robots need to be carefully considered before introducing robots to students as second language tutors. Additionally, in [52], incorporating robots into teacher education or professional development programs was proposed. Teachers, students, and social robots were said to become all key actors in future classrooms, and teachers' attitudes and beliefs were said to have possible influences on the future deployment of social robots.

- Care and assistance: in [61], the importance of experience in working with robots and raising awareness about what care robots can do was shown, in the objective of moving away from preconceptions based on fiction and imaginaries of care robots. In [58], an aim to achieve a design that is easily deployed in multiple locations, and contains all the necessary information for repeated deployments was expressed. In [205], an emotion recognition algorithm and an imitation algorithm was said to bring improvements to a robotic system for physical training of older adults. In [55], where a review of the usages of socially assistive robots in elderly care was made, the authors concluded that studies should be clearer about the precise role of any robot, and should use validated measures to assess their effectiveness. In the same context, a sophisticated speech analysis ability and accuracy of understanding language were said to be desired to improve the interaction between a human being and a robot in [59]. Under the topic of the acceptance of healthcare robots for older adults in [60], matching individual needs and preferences to the robot was said to possibly improve the acceptance. An alternatively proposed approach was to alter users' expectations to match the capabilities of the robot.

- Children companionship: in [41], it was said that humanoid robots are promising for robot-mediated education with primary school-aged children due to their capacity of making voices and gestures that motivate children in learning activities. In [79], where the work addressed robots playing games with people, it was said that a robot can have a sort of character that would support its perception as a rational agent, by taking measures such as adapting the behavior and strategy of the robot to the real-time perception it has of the humans it interacts with.

- Autism and medicine: in [77], a list of considerations to be taken into account when developing robots for children with ASD has been made. It shows that such robots should be user-focused, usable, reliable, safe, and affordable. In [74], the robotic scenario was said to be an excellent way to elicit behaviors in children with ASD through interaction, analysis of the child's behavior, and adaptation to it. According to the authors, Introducing robots into therapy would be of great clinical interest. In [75], works on social signal processing and socially assistive robotics were reported and issues that should be addressed by researchers in these research domains were listed. Among them are the machine understanding of typical and autistic behaviors, and the availability of databases of children with ASD interactions.

- security of robotic systems: an important aspect to address in social robotics is security and cybersecurity. Indeed, intelligent systems can help protect the security of users but hackers could attack social robot users from different vectors [206–208]. This should be taken into account when considering the use of social robots [209]. Work has been done in this domain to improve security, such as the guidelines published by a High-Level Expert Group established by the European Commission on trustworthy artificial intelligence. Additionally, the Trusted-ROS system was proposed in [207] for security improvement in humanoid robots. Additionally, recommendations were presented in [208] such as multi-factor authentication and multi-factor cryptography.

## 8. Conclusions

This paper showed a survey made on different aspects of social robotics, without a focus on a specific aspect. Applications of social robotics, modalities of the interaction of social robots with humans, platforms used, relationships with advances in different technological areas, and measures of acceptability of humans towards robots have been addressed. It can be seen that humanity can expect to co-exist with robots in the future, in different aspects of everyday life. This is not currently the case, with many of the previous work consisting of research studies rather than actual implementations of robots that are ready to use. This is not only due to the incomplete maturity of technologies that robots require to operate with humans, with enough performances to be trusted and believed, but also to the incomplete comprehension of how humans may be willing to accept robots and in which contexts. In this context, it was argued in [210] that there is a phenomenon becoming common in the social robotics field, named the roboid: a robot that is still at the prototype stage but claims to be fully functioning.

The survey made in this paper is not exhaustive with regards to any of the areas addressed but aims to provide readers with a tour of the world of social robotics. More areas can be addressed in the future, with similar surveys that can be done every few years, due to the rapid advance and expansion of social robotics technologies and usages.

**Author Contributions:** Conceptualization, K.Y., S.S., S.A. and T.B.; methodology, K.Y., S.S., S.A. and T.B.; software, K.Y. and S.S.; validation, S.S., S.A. and T.B.; formal analysis, K.Y.; investigation, K.Y. and S.S.; resources, K.Y., S.S., S.A. and T.B.; data curation, K.Y. and S.S.; writing—original draft preparation, K.Y. and S.S.; writing—review and editing, K.Y., S.S., S.A. and T.B.; visualization, K.Y. and S.S.; supervision, S.A. and T.B.; project administration, S.A. and T.B. All authors have read and agreed to the published version of the manuscript.

**Funding:** This research received no external funding.

**Institutional Review Board Statement:** Not applicable.

**Informed Consent Statement:** Not applicable.

**Conflicts of Interest:** The authors declare no conflict of interest.

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
