# Peer review of "A Survey on Recent Advances in Social Robotics"

_robotics, doi:10.3390/robotics11040075_

Round 1

Reviewer 1 Report

The article describes the trends in social robots across different disciplines. The authors provide a good breakdown of these robots with respect to disciplines, modalities and evaluation tools. While several examples of studies with social robots were described per discipline and modularity; each study offered different research goals and target populations that was hard to follow at times. I’d suggest to summarize these studies/robots in tables to describe the research goal, robot, end-users, and results similar to Table 1. In addition, the article should be proof-read for grammar mistakes and redundant sentences before publication. Other comments are below:

As a literature review of social robots, did the authors follow a systematic analysis such as PRISM-A? What was the inclusion criteria to select the cited articles? What was the academic/clinical background of those who selected the articles?

The description of Section 2.6. Children companionship seem to fit the Education Section. Maybe rephrase the description as it could also be considered as education from a social and spatial context.

Line 423-436: It was unclear what was the goal of describing the future work of research [88]. Does this fill a research gap in the field of social robots? Maybe rephrase these specifications towards the design of social robotics to improve its ‘conversation capabilities’.

Lines 488-515 seems redundant also is also described in Table 1.

Section 5.2.1. Missing heading?

Section 6. Metrics mention a few metrics and assessment tools to measure human perception and acceptability; however, the authors mentioned that these tools are subjective and results may vary with time. What are the implications of these results for the field of social robotics? What metrics could be used to address these limitations? Please elaborate.

Author Response

Dear Respected Reviewer,

The authors would like to take this opportunity to thank the reviewers for their time, consideration, and insightful feedback. All the reviewer’s comments and suggestions have been taken into consideration and addressed in the revised version of the paper. All changes have been highlighted in yellow in the revised manuscript.

A point to point report is attached as reply to reviewer#1

Reviewer 2 Report

The paper overviews recent advances in social robotics from different perspectives providing examples of future directions. The paper is interesting and addresses a topic of great interest for the research community. The paper is well structured and written.

However, I recommend authors to list the contributions of the proposed survey in the introduction and to give a title to section 5.2.1.

Author Response

Dear Respected Reviewer,

The authors would like to take this opportunity to thank the reviewers for their time, consideration, and insightful feedback. All the reviewer’s comments and suggestions have been taken into consideration and addressed in the revised version of the paper. All changes have been highlighted in yellow in the revised manuscript.

A point to point report is attached 

Reviewer 3 Report

In my opinion, an interesting article at a good level, corresponding to the current trends, knowledge and achievements in this area.

The presented material shows how much has been achieved in the field of social robots, but at the same time shows how much still needs to be done - it will require many years of work of researchers and research centers, and this research should undoubtedly be carried out, because robots are and will be among us! - that's why I believe that this social aspect is extremely important for each party.

The authors drew attention to many important aspects, although I would like to draw attention to (of a discussion nature for the future), for example: what about security / cybersecurity?

Nevertheless, I consider the work valuable and recommend it for publication.

Author Response

Dear Respected Reviewer,

The authors would like to take this opportunity to thank the reviewers for their time, consideration, and insightful feedback. All the reviewer’s comments and suggestions have been taken into consideration and addressed in the revised version of the paper. All changes have been highlighted in yellow in the revised manuscript.

A point to point report is attached as reply to reviewer#2
